# Lifelong Learning with Weighted Majority Votes

**Anastasia Pentina**
IST Austria
apentina@ist.ac.at

**Ruth Urner**
Max Planck Institute for Intelligent Systems
rurner@tuebingen.mpg.de

## Abstract

Better understanding of the potential benefits of information transfer and repre-
sentation learning is an important step towards the goal of building intelligent
systems that are able to persist in the world and learn over time. In this work, we
consider a setting where the learner encounters a stream of tasks but is able to retain
only limited information from each encountered task, such as a learned predictor.
In contrast to most previous works analyzing this scenario, we do not make any
distributional assumptions on the task generating process. Instead, we formulate a
complexity measure that captures the diversity of the observed tasks. We provide a
lifelong learning algorithm with error guarantees for every observed task (rather
than on average). We show sample complexity reductions in comparison to solv-
ing every task in isolation in terms of our task complexity measure. Further, our
algorithmic framework can naturally be viewed as learning a representation from
encountered tasks with a neural network.

## 1 Introduction

Machine learning has made significant progress in understanding both theoretical and practical
aspects of solving a single prediction problem from a set of annotated examples. However, if we
aim at building autonomous agents, capable to persist in the world, we need to establish methods
for continuously learning various tasks over time [25, 26]. There is no hope to initially provide, for
example, an autonomous robot with sufficiently rich prior knowledge to solve any problem that it may
encounter during the course of its life. Therefore, an important goal of machine learning research
is to replicate humans' ability to learn from experience and to reuse knowledge from previously
encountered tasks for solving new ones more efficiently. This is aimed at in *lifelong learning* or
*learning to learn*, where a learning algorithm is assumed to encounter a stream of tasks and is aiming
to exploit commonalities between them by transferring information from earlier tasks to later ones.

The first theoretical formulation of this framework was proposed by Baxter [4]. In that model, tasks
are generated by a probability distribution and the goal, given a sample of tasks from this distribution,
is to perform well in expectation over tasks. Under certain assumptions, such as a shared good
hypothesis set, this model allows for sample complexity savings [4]. However, good performance in
expectation is often too weak a requirement. To stay with the robot example, failure on a single task
may cause severe malfunction – and the end of the robot's life. Moreover, the theoretical analysis of
this model relies on the assumption that the learner maintains access to training data for all previously
observed tasks, which allows to formulate a joint optimization problem. However, it is unlikely that
an autonomous robot is able to keep all this data.

Thus, we instead focus on a streaming setting for lifelong learning, where the learner can only
retain learned models from previously encountered tasks. These models have a much more compact
description than the joint training data. Specifically, we are interested in analysis and performance
guarantees in the scenario that 1) tasks arrive one at a time without distributional or i.i.d. assumptions,
2) the learner can only keep the learned hypotheses from previously observed tasks, 3) error bounds
are required for every single task, rather than on average.

The first analysis of this challenging setting was recently provided by Balcan *et al.* [3]. That work demonstrates sample complexity improvements for learning linear halfspaces (and some boolean function classes) in the lifelong learning setting in comparison to solving each task in isolation under the assumption that the tasks share a common low dimensional representation. However, the analysis relies on the marginal distributions of all tasks being isotropic log-concave. It was stated as an open challenge in that work whether similar guarantees (error bounds for every task, while only keeping limited information from earlier tasks) were possible under less restrictive distributional assumptions.

In this work, we (partially) answer this question in the positive. We do so by proposing to learn with *weighted majority votes rather than linear combinations* over linear predictors. We show that the shift from linear combinations to majority votes introduces stability to the learned ensemble that allows exploiting it for later tasks. Additionally, we show that this stability is achieved for *any ground hypothesis class*. We formulate a relatedness assumption on the sequence of tasks (similar to one used in [3]) that captures how suitable to lifelong learning a sequence of tasks is. With this, we prove that sample complexity savings through lifelong learning are obtained *for arbitrary marginal distributions* (provided that these marginal distributions are related in terms of their discrepancy [5, 17]). This is a significant generalization towards more practically relevant scenarios.

**Summary of our work**   We employ a natural algorithmic paradigm, similar to the one in [3]. The algorithm maintains a set of *base hypotheses* from some fixed *ground* hypothesis class $\mathcal{H}$. These base hypotheses are predictors learned on previous tasks. For each new task, the algorithm first attempts to achieve good prediction performance with a weighted majority vote over the current base hypotheses, and uses this predictor for the task if successful. Otherwise (if no majority vote classifier achieves high accuracy), the algorithm resorts to learning a classifier from the ground class for this task. This classifier is then added to the set of base hypotheses, to be used for subsequent tasks. We describe this algorithm Section 4.1.

If the ground class is the class of linear predictors, this algorithm is actually learning a neural network. Each base classifier becomes a node in a hidden middle layer, which represents a learned feature representation of the neural net. A new task is then either solved by employing the representation learned from previous tasks (the current middle layer), and just learning task specific weights for the last layer, or, in case this is not possible, it extends the current representation. See also Section 4.2.

This paradigm yields sample complexity savings, if the tasks encountered are related in the sense that for many tasks, good classification accuracy can be achieved with a weighted majority vote over previously learned models. We formally capture this property as an *effective dimension* of the sequence of tasks. We prove in Section 4.3 that if this effective dimension is bounded by $k$, then the total sample complexity for learning $n$ tasks with this paradigm is upper bounded by $\tilde{O}\left((nk + \mathrm{VC}(\mathcal{H})k^2)/\epsilon\right)$, a reduction from $\tilde{O}\left(n\mathrm{VC}(\mathcal{H})/\epsilon\right)$, the sample complexity of learning $n$ tasks individually without retaining information from previous tasks.

The main technical difficulty is to control the propagation of errors. Since every task is represented by a finite training set, the learner has access only to approximations of the true labeling functions, which may degrade the quality of this collection of functions as a "basis" for new tasks. Balcan *et al.* [3] control this error propagation using elegant geometric arguments for linear predictors under isotropic log-concave distributions. We show that moving from linear combinations to majority votes yields the required stability for the quality of the representation under arbitrary distributions.

Finally, while we first present our algorithm and results for known upper bounds of $k$ base tasks and $n$ tasks in total, we also provide a variation of the algorithm that does not need to know the number of tasks and the complexity parameter of the task sequence. We show that similar sample complexity improvements are achievable in this setting in Section 5.

## 2   Related Work

**Lifelong learning.**   While there are many ways in which prediction tasks may be related [20], most of the existing approaches to transfer or lifelong learning are exploiting possible similarities between the optimal predictors for the considered tasks. In particular, one widely used relatedness assumption is that these predictors can be described as linear or sparse combinations of some common meta-features and the corresponding methods aim at learning this representations [10, 2, 15, 3]. Though this idea was originally used in the multi-task setting, it was later extended to lifelong learning by

Eaton *et al.* [9], who proposed a method for sequentially updating the underlying representation as new tasks arrive. These settings were theoretically analyzed in a series of works [18, 19, 23, 21] that have demonstrated that information transfer can lead to provable sample complexity reductions compared to solving each task independently. However, all these results rely on Baxter's model of lifelong learning and therefore assume access to the training data for all (observed) tasks and provide guarantees only on average performance over all tasks. An exception is [6], where the authors provide error guarantees for every task in the multi-task scenario. However, these guarantees are due to the relatedness assumption used which implies that all tasks have the same expected error. The task relatedness assumption that we employ is related to the one used in [1] for multi-task learning with expert advice. There the authors consider a setting where there exists a small subset of experts that perform well on all tasks. Similarly, we assume that there is a small subset of base tasks, such that the remaining ones can be solved well using majority votes over the corresponding base hypotheses.

**Majority votes.**   Weighted majority votes are a theoretically well understood and widely used in practice type of ensemble predictor. In particular, they are employed in boosting [11]. They are also often considered in works that utilize PAC-Bayesian techniques [22, 12]. Majority votes are also used in the concept drift setting [14]. The corresponding method, conceptually similar to the one proposed here, dynamically updates a set of experts and uses their weighted majority votes for making predictions.

## 3   Formal Setting

### 3.1   General notation and background

We let $\mathcal{X} \subseteq \mathbb{R}^d$ denote a *domain* set and let $\mathcal{Y}$ denote a *label* set. A *hypothesis* is a function $h : \mathcal{X} \to \mathcal{Y}$, and a *hypothesis class* $\mathcal{H}$ is a set of hypotheses. We model learning tasks as pairs $\langle D, h^* \rangle$ of a distribution $D$ over $\mathcal{X}$ and a labeling function $h^* : \mathcal{X} \to \mathcal{Y}$. The quality of a hypothesis is measured by a *loss function* $\ell : \mathcal{Y} \times \mathcal{Y} \to \mathbb{R}^+$. We deal with binary classification tasks, that is, $\mathcal{Y} = \{-1, 1\}$, under the 0/1-loss function, that is, $\ell(y, y') = [\![y \neq y']\!]$ (we let $[\![\cdot]\!]$ denote the indicator function). The *risk* of a hypothesis $h$ with respect to task $\langle D, h^* \rangle$ is defined as its expected loss:

$$\mathcal{L}_{D,h^*}(h) := \mathbb{E}_{x \sim D}[\ell(h(x), h^*(x))].$$

Given a sample $S = \{(x_1, y_1), (x_2, y_2), \ldots, (x_n, y_n)\}$, the *empirical risk* of $h$ with respect to $S$ is

$$\mathcal{L}_S(h) := \frac{1}{n} \sum_{i=1}^{n} \ell(h(x_i), y_i).$$

For binary classification, the sample complexity of learning a hypothesis class is characterized (that is, upper and lower bounded), by the VC-dimension of the class [27]. We will employ the following generalization bounds for classes of finite VC-dimension:

**Theorem 1** (Corollaries 5.2 and 5.3 in [7]). *Let $\mathcal{H}$ be a class of binary functions with a finite VC-dimension. There exists a constant $C$, such that for any $\delta \in (0, 1)$, for any task $\langle D, h^* \rangle$, with probability at least $1 - \delta$ over a training set $S$ of size $n$, sampled i.i.d. from $\langle D, h^* \rangle$:*

$$\mathcal{L}_{D,h^*}(\hat{h}) \leq \mathcal{L}_S(\hat{h}) + \sqrt{\mathcal{L}_S(\hat{h}) \cdot \Delta} + \Delta, \tag{1}$$

$$\mathcal{L}_S(\hat{h}) \leq \mathcal{L}_{D,h^*}(\hat{h}) + \sqrt{\mathcal{L}_{D,h^*}(\hat{h}) \cdot \Delta} + \Delta, \tag{2}$$

$$\mathcal{L}_{D,h^*}(\hat{h}) \leq \inf_{h \in \mathcal{H}} \mathcal{L}_D(h) + \sqrt{\inf_{h \in \mathcal{H}} \mathcal{L}_D(h) \cdot \Delta} + \Delta, \tag{3}$$

*where $\hat{h} \in \arg\min_{h \in \mathcal{H}} \mathcal{L}_S(h)$ is an empirical risk minimizer and*

$$\Delta = C \frac{\text{VC}(\mathcal{H}) \log(n) + \log(1/\delta)}{n}. \tag{4}$$

In the realizable case ($h^* \in \mathcal{H}$), the above bounds imply that the sample complexity is upper bounded by $\tilde{O}\left(\frac{\text{VC}(\mathcal{H}) + \log(1/\delta)}{\epsilon}\right)$.

**Weighted majority votes** Given a hypothesis class $\mathcal{H}$, we define the class of $k$-majority votes as

$$\mathrm{MV}(\mathcal{H}, k) = \left\{ g : \mathcal{X} \to \mathcal{Y} \mid \exists h_1, \ldots, h_k \in \mathcal{H}, \exists w_1, \ldots, w_k \in \mathbb{R} : \; g(x) = \mathrm{sign}\left(\sum_{i=1}^{k} w_i h_i(x)\right) \right\}.$$

We will omit $k$ in the above notation if clear from the context. The VC-dimension of $\mathrm{MV}(\mathcal{H}, k)$ is upper bounded as

$$\mathrm{VC}(\mathrm{MV}(\mathcal{H}, k)) \leq k(\mathrm{VC}(\mathcal{H}) + 1)(3 \log(\mathrm{VC}(\mathcal{H}) + 1) + 2) \tag{5}$$

(see Theorem 10.3 in [24]). This implies in particular, that the VC-dimension of majority votes over a fixed set of $k$ functions is upper bounded by $\tilde{O}(k \log(k))$.

## 3.2 Lifelong learning

In the lifelong learning setting the learner encounters a stream of prediction problems $\langle D_1, h_1^* \rangle, \ldots, \langle D_n, h_n^* \rangle$, one at a time. In this work we focus on the realizable case, i.e. $h_i^* \in \mathcal{H}$ for every $i$ and some fixed $\mathcal{H}$. In contrast to most works on lifelong learning, we assume that the only information the learner is able to store about the already observed tasks is the obtained predictors, i.e. it does not have access to the training data of already solved tasks. The need to keep little information from previous tasks has also been argued for in the context of domain adaptation [16].

Possible benefits of information transfer depend on how related or, in other words, how diverse the observed tasks are. Moreover, since we do not make any assumptions on the task generation process (in contrast to Baxter's i.i.d. model [4]), we will formulate our relatedness assumption in terms of a sequence of tasks. Intuitively, one would expect that the information transfer is beneficial if only a few times throughout the course of learning information obtained from the already solved tasks will not be sufficient to solve the current one. In order to formalize this intuition, we use the following (pseudo-)metric over the hypothesis class with respect to a marginal distribution $D$:

$$d_D(h, h') = \mathop{\mathbb{E}}_{x \sim D} [\![ h(x) \neq h'(x) ]\!]. \tag{6}$$

Further, we can define a distance of a hypothesis to a hypothesis space as

$$d_D(h, \mathcal{H}') = \min_{h' \in \mathcal{H}'} d_D(h, h') \tag{7}$$

and the distance between two sets of hypotheses as

$$d_D(\mathcal{H}, \mathcal{H}') = \max_{h \in \mathcal{H}} d_D(h, \mathcal{H}') = \max_{h \in \mathcal{H}} \min_{h' \in \mathcal{H}'} d_D(h, h'). \tag{8}$$

Note that the latter is not necessarily a metric over subsets of the hypothesis space. However, it does satisfy the triangle inequality (see Section 1 in the supplementary material).

Now we can formulate the diversity measure for a sequence of learning tasks that we will employ. Note that the concepts below are closely related to the ones used in [3] for the case of linear predictors and linear combinations over these.

**Definition 1.** *A sequence of learning tasks* $\langle D_1, h_1^* \rangle, \ldots, \langle D_n, h_n^* \rangle$ *is* $\gamma$-*separated, if for every* $i$ $d_{D_i}(h_i^*, \mathrm{MV}(h_1^*, \ldots, h_{i-1}^*)) > \gamma$.

**Definition 2.** *A sequence of learning tasks* $\langle D_1, h_1^* \rangle, \ldots, \langle D_n, h_n^* \rangle$ *has* $\gamma$-*effective dimension* $k$, *if the largest* $\gamma$-*separated subsequence of these tasks has length* $k$.

Formally, we will assume that the $\gamma$-effective dimension $k$ of the observed sequence of tasks is relatively small for a sufficiently small $\gamma$. Note that this assumption can also be seen as a relaxation of the one used in [8]. There the authors assumed that there exists a set of $k$ hypothesis such that every task can be well solved by one of them. This would correspond to substituting the sets of weighted majority votes $\mathrm{MV}(h_1^*, \ldots, h_{i-1}^*)$ by just the collections $\{h_1^*, \ldots, h_{i-1}^*\}$ in the above definitions.

Moreover, we will assume that the marginal distributions have small *discrepancy* with respect to the hypothesis set $\mathcal{H}$:

$$\mathrm{disc}_{\mathcal{H}}(D_i, D_j) = \max_{h, h' \in \mathcal{H}} |d_{D_i}(h, h') - d_{D_j}(h, h')|. \tag{9}$$

This is a measure of task relatedness that has been introduced in [13] and shown to be beneficial in the context of domain adaptation [5, 17]. Note, however, that we do not make any assumptions on the marginal distributions $D_1, \ldots, D_n$ themselves.

# 4 Algorithm and complexity guarantees

## 4.1 The algorithm

We employ a natural algorithmic paradigm, which is similar to the one in [3]. Algorithm 1 below provides pseudocode for our procedure. The algorithm takes as parameters a class $\mathcal{H}$, which we call the *ground class*, accuracy and confidence parameters $\epsilon$ and $\delta$, as well as a task horizon $n$ (the number of tasks to be solved) and a parameter $k$ (a guessed upper bound on the number of tasks that will not be solvable as majority votes over earlier tasks). In Section 5 we present a version that does not need to know $n$ and $k$ in advance.

---

**Algorithm 1** Lifelong learning of majority votes

---

1: **Input** parameters $\mathcal{H}, n, k, \epsilon, \delta$
2: set $\delta' = \delta/(2n)$, $\epsilon' = \epsilon/(8k)$
3: draw a training set $S_1$ from $\langle D_1, h_1^* \rangle$, such that $\Delta_1 := \Delta(\mathrm{VC}(\mathcal{H}), \delta', |S_1|) \leq \epsilon'$
4: $g_1 = \arg\min_{h \in \mathcal{H}} \mathcal{L}_{S_1}(h)$
5: set $\tilde{k} = 1, i_1 = 1, \tilde{h}_1 = g_1$
6: **for** $i = 2$ to $n$ **do**
7: $\quad$ draw a training set $S_i$ from $\langle D_i, h_i^* \rangle$, such that $\Delta_i := \Delta(\mathrm{VC}(\mathrm{MV}(\tilde{h}_1, \ldots, \tilde{h}_{\tilde{k}})), \delta', |S_i|) \leq \frac{\epsilon}{40}$
8: $\quad$ $g_i = \arg\min_{h \in \mathrm{MV}(\tilde{h}_1, \ldots, \tilde{h}_{\tilde{k}})} \mathcal{L}_{S_i}(h)$
9: $\quad$ **if** $\mathcal{L}_{S_i}(g_i) + \sqrt{\mathcal{L}_{S_i}(g_i) \cdot \Delta_i} + \Delta_i > \epsilon$ **then**
10: $\quad\quad$ draw a training set $S_i$ from $\langle D_i, h_i^* \rangle$, such that $\Delta_i := \Delta(\mathrm{VC}(\mathcal{H}), \delta', |S_i|) \leq \epsilon'$
11: $\quad\quad$ $g_i = \arg\min_{h \in \mathcal{H}} \mathcal{L}_{S_i}(h)$
12: $\quad\quad$ set $\tilde{k} = \tilde{k} + 1, \tilde{h}_{\tilde{k}} = g_i, i_{\tilde{k}} = i$
13: $\quad$ **end if**
14: **end for**
15: **return** $g_1, \ldots, g_n$

---

During the course of its "life", the algorithm maintains a set of *base hypotheses* $(\tilde{h}_1, \ldots, \tilde{h}_{\tilde{k}})$ from the ground class, which are predictors learned on previous tasks. In order to solve the first task, it uses the hypothesis set $\mathcal{H}$ and a large enough training set $S_1$ to ensure the error guarantee $\epsilon' \leq \epsilon/8k$ with probability at least $1 - \delta'$, where $\delta' = \delta/2n$. The learned hypothesis $\tilde{h}_1 \in \mathcal{H}$ is the first member of the set of base hypotheses. For each new task $i$, the algorithm first attempts to achieve good prediction performance (up to error $\epsilon$) with a weighted majority vote over the base hypotheses, i.e. it attempts to learn this task using the class $\mathrm{MV}(\tilde{h}_1, \ldots, \tilde{h}_{\tilde{k}})$, and uses the obtained predictor for the task if successful. Otherwise (if no majority vote classifier achieves high accuracy), the algorithm resorts to learning a classifier from the base class for this task, which is then added to the set of base hypotheses, to be used for subsequent tasks. The error guarantees are ensured with Theorem 1 by choosing the training sets $S_i$ large enough so that

$$\Delta_i := \Delta(\mathrm{VC}(\mathcal{H}_i), \delta', |S_i|) := C \frac{\mathrm{VC}(\mathcal{H}_i)\log(|S_i|) + \log(1/\delta')}{|S_i|} \leq c\epsilon,$$

where $\mathcal{H}_i$ is either the ground class $\mathcal{H}$ or the set of weighted majority votes over the current set of base hypotheses $\mathrm{MV}(\tilde{h}_1, \ldots, \tilde{h}_{\tilde{k}})$, and constant $c$ is set according to case, see pseudocode.

While this approach is very natural, the challenge is to analyze it and to specify the parameters. In particular, we need to ensure that the algorithm will not have to search over (potentially large) hypothesis set $\mathcal{H}$ too often and, consequently, will lead to provable sample complexity reductions over solving each task independently. The following theorem summarizes the performance guarantees for Algorithm 1 (the proof is in Section 4.3).

**Theorem 2.** *Consider running Algorithm 1 on a sequence of tasks with $\gamma$-effective dimension at most $k$ and $\mathrm{disc}_{\mathcal{H}}(D_i, D_j) \leq \xi$ for all $i, j$. Then, if $\gamma \leq \epsilon/4$ and $k\xi < \epsilon/8$, with probability at least $1 - \delta$:*

- *The error of every task is bounded: $\mathcal{L}_{D_i, h_i^*}(g_i) \leq \epsilon$ for every $i = 1, \ldots, n$.*

- *The total number of labeled examples used is $\tilde{O}\left(\frac{nk + \mathrm{VC}(\mathcal{H})k^2}{\epsilon}\right)$.*

**Discussion** Note that if we assume that all tasks are realizable by $\mathcal{H}$, independently learning them up to error $\epsilon$ would have sample complexity $\tilde{O}\left(\frac{\mathrm{VC}(\mathcal{H})n}{\epsilon}\right)$. The sample complexity of learning $n$ tasks in the lifelong learning regime with our paradigm in contrast is $\tilde{O}\left(\frac{nk+\mathrm{VC}(\mathcal{H})k^2}{\epsilon}\right)$. This is a significant reduction if the effective dimension of the task sequence $k$ is small in comparison to the total number $n$ of tasks, as well as the complexity measure $\mathrm{VC}(\mathcal{H})$ of the ground class. That is, if most tasks are learnable as combination of previously stored base predictors, much less data is required overall.

Note that for all those tasks that are solved as majority votes, our algorithm and analysis actually require realizability only by the class of $k$-majority votes over $\mathcal{H}$ and not by the ground class $\mathcal{H}$. Learning the $n$ tasks independently under this assumption, has sample complexity $\tilde{O}\left(\frac{\mathrm{VC}(\mathcal{H})k}{\epsilon} + \frac{(n-k)\mathrm{VC}(\mathcal{H})k}{\epsilon}\right)$. In contrast, the lifelong learning method gradually identifies the relevant set of base predictors and thereby reduces the number of required examples.

## 4.2 Neural networks

If the ground class is the class of linear predictors, our algorithm is actually learning a neural network (with $\mathrm{sign}()$ as the activation function). Each base classifier becomes a new node in a hidden middle layer. Thus, the maintained set of base classifiers can be viewed as feature representation in the neural net, which was learned based on the encountered tasks. A new task is then either solved by employing the representation learned from previous tasks (the current middle layer), and just learning task specific weights for the last layer; or, in case this is not possible, a fresh linear classifier is learned, and added as a node to the middle layer. Thus, in this case, the feature representation is extended.

## 4.3 Analysis

We start with presenting the following two lemmas that show how to control the error propagation of the learned representations (sets of base classifiers). We then proceed to the proof of Theorem 2.

**Lemma 1.** *Let* $V = \mathrm{MV}(h_1, \ldots, h_k, g)$ *and* $\tilde{V} = \mathrm{MV}(h_1, \ldots, h_k, \tilde{g})$. *Then, for any distribution $D$:*

$$d_D(V, \tilde{V}) \leq d_D(g, \tilde{g}). \tag{10}$$

*Proof.* By the definition of $d_D(V, \tilde{V})$ there exists $u \in V$ such that:

$$d_D(V, \tilde{V}) = d_D(u, \tilde{V}). \tag{11}$$

We can represent $u$ as $u = \mathrm{sign}(\sum_{i=1}^{k} \alpha_i h_i + \alpha g)$ and let $u_1 = \sum_{i=1}^{k} \alpha_i h_i$. Note that while all $h_i$-s, $g$ and $\tilde{g}$ are assumed to take values in $\{-1, 1\}$, $u_1$ can take values in $\mathbb{R}$. Then:

$$
\begin{aligned}
d_D(u, \tilde{V}) &= \min_{\tilde{h} \in \tilde{V}} d_D(u, \tilde{h}) \leq \min_{\tilde{h} \in \mathrm{MV}(u_1, \tilde{g})} d_D(u, \tilde{h}) \\
&\leq \max_{h \in \mathrm{MV}(u_1, g)} \min_{\tilde{h} \in \mathrm{MV}(u_1, \tilde{g})} d_D(h, \tilde{h}) = d_D(\mathrm{MV}(u_1, g), \mathrm{MV}(u_1, \tilde{g})).
\end{aligned}
$$

Now we show that for any $\alpha_1 u_1 + \alpha_2 g \in \mathrm{MV}(u_1, g)$ there exists a close hypothesis in $\mathrm{MV}(u_1, \tilde{g})$. In particular, this hypothesis is $\alpha_1 u_1 + \alpha_2 \tilde{g}$:

$$
\begin{aligned}
d_D(\alpha_1 u_1 + \alpha_2 g, \alpha_1 u_1 + \alpha_2 \tilde{g}) &= \mathop{\mathbb{E}}_{x \sim D} [\![ \mathrm{sign}(\alpha_1 u_1(x) + \alpha_2 g(x)) \neq \mathrm{sign}(\alpha_1 u_1(x) + \alpha_2 \tilde{g}(x)) ]\!] \\
&= \mathop{\mathbb{E}}_{x \sim D} [\![ \alpha_1^2 u_1^2(x) + \alpha_1 \alpha_2 u_1(x) g(x) + \alpha_1 \alpha_2 u_1(x) \tilde{g}(x) + \alpha_2^2 g(x) \tilde{g}(x) < 0 ]\!].
\end{aligned}
$$

Note that for every $x$ on which $g$ and $\tilde{g}$ agree, i.e. $g(x)\tilde{g}(x) = 1$, we obtain:

$$\alpha_1^2 u_1^2(x) + \alpha_1 \alpha_2 u_1(x) g(x) + \alpha_1 \alpha_2 u_1(x) \tilde{g}(x) + \alpha_2^2 g(x) \tilde{g}(x) = (\alpha_1 u_1(x) + \alpha_2 g(x))^2 \geq 0.$$

Therefore:

$$d_D(\alpha_1 u_1 + \alpha_2 g, \alpha_1 u_1 + \alpha_2 \tilde{g}) \leq \mathop{\mathbb{E}}_{x \sim D} [\![ g(x) \neq \tilde{g}(x) ]\!] = d_D(g, \tilde{g}). \tag{12}$$

$\square$

**Lemma 2.** *Let* $V_k = \mathrm{MV}(h_1, \ldots, h_k)$ *and* $\tilde{V}_k = \mathrm{MV}(\tilde{h}_1, \ldots, \tilde{h}_k)$. *For any distribution* $D$, *if* $d_D(h_i, \tilde{h}_i) \leq \epsilon_i$ *for every* $i = 1, \ldots, k$, *then* $d_D(V_k, \tilde{V}_k) \leq \sum_{i=1}^{k} \epsilon_i$.

For the proof see Section 2 in the supplementary material.

*Proof of Theorem 2.* 1. First, note that for every task Algorithm 1 solves at most 2 estimation problems with a probability of failure $\delta'$ for each of them. Therefore, with a union bound argument, the probability of any of these estimations being wrong is at most $2 \cdot n \cdot \delta' = \delta$. Thus, from now we assume that all the estimations were correct, that is, the high probability events of Theorem 1 hold.

2. To see that the error of every encountered task is bounded by $\epsilon$, note that there are two cases. For tasks $i$ that are solved by a majority vote over previous tasks, we have $\mathcal{L}_{S_i}(g_i) + \sqrt{\mathcal{L}_{S_i}(g_i) \cdot \Delta_i} + \Delta_i \leq \epsilon$. In this case, Equation (1) in Theorem 1 implies $\mathcal{L}_{D_i, h_i^*}(g_i) \leq \epsilon$. For tasks $i$ that are not solved as a majority vote over previous tasks, we have $\Delta_i = \Delta(\mathrm{VC}(\mathrm{MV}(\tilde{h}_1, \ldots, \tilde{h}_{\tilde{k}})), \delta', m) \leq \epsilon/8k$. Since task $i$ is realizable by the base class $\mathcal{H}$, we have $\inf_{h \in \mathcal{H}} \mathcal{L}_{D_i, h_i^*}(h) = 0$, and thus Equation (3) of Theorem 1 implies $\mathcal{L}_{D_i, h_i^*}(g_i) \leq \epsilon/8k < \epsilon$.

3. To upper bound the sample complexity we first prove that the number $\tilde{k}$ of tasks, which are not learned as majority votes over previous tasks, is at most $k$. For that we use induction showing that for every $\hat{k} \leq \tilde{k}$, when we create a new $\tilde{h}_{\hat{k}}$ from the $i_{\hat{k}}$-th task, we have that

$$d_{D_{i_{\hat{k}}}}(h_{i_{\hat{k}}}^*, \mathrm{MV}(h_{i_1}^*, \ldots, h_{i_{\hat{k}-1}}^*)) > \gamma. \tag{13}$$

This implies $\tilde{k} \leq k$ by invoking that the $\gamma$-effective dimension of the sequence of encountered tasks is at most $k$.

To proceed to the induction, note that for $\hat{k} = 1$, the claim follows immediately. Consider $\hat{k} > 1$. If we create a new $\tilde{h}_{\hat{k}}$, it means that the condition in line 9 is true, which is:

$$\mathcal{L}_{S_{i_{\hat{k}}}}(g_{i_{\hat{k}}}) + \sqrt{\mathcal{L}_{S_{i_{\hat{k}}}}(g_{i_{\hat{k}}}) \cdot \Delta_i} + \Delta_i > \epsilon. \tag{14}$$

Therefore $\mathcal{L}_{S_{i_{\hat{k}}}}(g_{i_{\hat{k}}}) > 0.83\epsilon$. Consequently, due to (2), $\mathcal{L}_{D_{i_{\hat{k}}}, h_{i_{\hat{k}}}^*}(g_{i_{\hat{k}}}) > 0.67\epsilon$. Finally, by (3), $\inf_g \mathcal{L}_{D_{i_{\hat{k}}}, h_{i_{\hat{k}}}^*}(g) > 0.5\epsilon$. Therefore there is no majority vote predictor based on $\tilde{h}_1, \ldots, \tilde{h}_{\hat{k}-1}$ that leads to error less than $\epsilon/2$ on the problem $i_{\hat{k}}$. In other words:

$$d_{D_{i_{\hat{k}}}}(h_{i_{\hat{k}}}^*, \mathrm{MV}(\tilde{h}_1, \ldots, \tilde{h}_{\hat{k}-1})) > \epsilon/2. \tag{15}$$

Now, by way of contradiction, suppose that $d_{D_{i_{\hat{k}}}}(h_{i_{\hat{k}}}^*, \mathrm{MV}(h_{i_1}^*, \ldots, h_{i_{\hat{k}-1}}^*)) \leq \gamma$. By construction for every $j = 1, \ldots, \hat{k} - 1$ $d_{D_{i_j}}(h_{i_j}^*, \tilde{h}_j) \leq \epsilon' \leq \epsilon/8k$. By the definition of discrepancy and the assumption on the marginal distributions it follows that for all $j$:

$$d_{D_{i_{\hat{k}}}}(h_{i_j}^*, \tilde{h}_j) \leq d_{D_{i_j}}(h_{i_j}^*, \tilde{h}_j) + \mathrm{disc}_{\mathcal{H}}(D_{i_j}, D_{i_{\hat{k}}}) \leq \epsilon' + \xi. \tag{16}$$

Therefore by Lemma 2:

$$d_{D_{i_{\hat{k}}}}(\mathrm{MV}(h_{i_1}^*, \ldots, h_{i_{\hat{k}-1}}^*), \mathrm{MV}(\tilde{h}_1, \ldots, \tilde{h}_{\hat{k}})) \leq k(\epsilon' + \xi). \tag{17}$$

Consequently, by using the triangle inequality:

$$d_{D_{i_{\hat{k}}}}(h_{i_{\hat{k}}}^*, \mathrm{MV}(\tilde{h}_1, \ldots, \tilde{h}_{\hat{k}-1})) \leq \gamma + k(\epsilon' + \xi) \leq \epsilon/4 + \epsilon/8 + \epsilon/8 = \epsilon/2, \tag{18}$$

which is in contradiction with (15).

4. The total sample complexity of Algorithm 1 consists of two parts. First, for every task Algorithm 1 checks, whether it can be solved by a majority vote over the base, at most $\tilde{k}$ predictors. For that it employs Theorem 1 and therefore needs the following number of samples:

$$\tilde{O}\left(\frac{n\tilde{k}\log\tilde{k}\log(\tilde{k}\log\tilde{k})\log(2n/\delta)}{\epsilon}\right) = \tilde{O}\left(\frac{nk}{\epsilon}\right). \tag{19}$$

Second, there are at most $\tilde{k}$ tasks that satisfy the condition in line 9 and are learned using the hypothesis set $\mathcal{H}$ with estimation error $\epsilon' = \epsilon/(8k)$. Therefore the corresponding sample complexity is: $O\left(\frac{\tilde{k}\mathrm{VC}(\mathcal{H})\log(2n/\delta)}{\epsilon/(8k)}\right) = \tilde{O}\left(\frac{\mathrm{VC}(\mathcal{H})k^2}{\epsilon}\right)$.  □

# 5 Lifelong learning with unknown horizon

In this section we present a modification of Algorithm 1 for the case when the total number of tasks $n$ and the complexity of the task sequence $k$ are not known in advance. The main difference between Algorithm 2 and Algorithm 1 is that with unknown $n$ and $k$ the learner has to adopt the parameters $\delta'$ and $\epsilon'$ on the fly. We show that this can be done by the doubling trick that is often used in online learning. Theorem 3 summarizes the resulting guarantees (the proof can be found in the supplementary material, Section 3).

---

**Algorithm 2** Lifelong learning of majority votes with unkown horizon

---
1: **Input** parameters $\mathcal{H}, \epsilon, \delta$
2: set $\delta_1 = \delta/2$, $\epsilon'_1 = \epsilon/16$
3: draw a training set $S_1$ from $\langle D_1, h_1^* \rangle$ of size $m$, such that $\Delta(\mathrm{VC}(\mathcal{H}), \delta_1, m) \leq \epsilon'_1$ (see (4))
4: $g_1 = \arg\min_{h \in \mathcal{H}} \mathcal{L}_{S_1}(h)$
5: set $\tilde{k} = 1$, $i_1 = 1$, $\tilde{h}_1 = g_1$
6: **for** $i = 2$ to $n$ **do**
7:     set $l = \lfloor \log i \rfloor$, $m = \lfloor \log \tilde{k} + 1 \rfloor$
8:     set $\delta_i = \frac{\delta}{2^{2l+2}}$, $\epsilon'_i = \frac{\epsilon}{2^{2m+4}}$
9:     draw a training set $S_i$ from $\langle D_i, h_i^* \rangle$ of size $m$, such that $\Delta(\mathrm{VC}(\mathrm{MV}(\tilde{h}_1, \ldots, \tilde{h}_{\tilde{k}})), \delta_i, m) \leq \epsilon/40$ (see (4))
10:     $g_i = \arg\min_{h \in \mathrm{MV}(\tilde{h}_1, \ldots, \tilde{h}_{\tilde{k}})} \mathcal{L}_{S_i}(h)$
11:     **if** $\mathcal{L}_{S_i}(g_i) + \sqrt{\mathcal{L}_{S_i}(g_i) \cdot \Delta} + \Delta > \epsilon$ **then**
12:         draw a training set $S_i$ from $\langle D_i, h_i^* \rangle$ of size $m$, such that $\Delta(\mathrm{VC}(\mathcal{H}), \delta_i, m) \leq \epsilon'_i$ (see (4))
13:         $g_i = \arg\min_{h \in \mathcal{H}} \mathcal{L}_{S_i}(h)$
14:         set $\tilde{k} = \tilde{k} + 1$, $\tilde{h}_{\tilde{k}} = g_i$, $i_{\tilde{k}} = i$
15:     **end if**
16: **end for**
17: **return** $g_1, \ldots, g_n$

---

**Theorem 3.** *Consider running Algorithm 2 on a sequence of tasks with $\gamma$-effective dimension at most $k$ and $\mathrm{disc}_{\mathcal{H}}(D_i, D_j) \leq \xi$ for all $i, j$. Then, if $\gamma \leq \epsilon/4$ and $k\xi < \epsilon/8$, with probability at least $1 - \delta$:*

- *The error of every task is bounded: $\mathcal{L}_{D_i, h_i^*}(g_i) \leq \epsilon$ for every $i = 1, \ldots, n$.*

- *The total number of labeled examples used is $\tilde{O}\left( \frac{nk + \mathrm{VC}(\mathcal{H})k^3}{\epsilon} \right)$.*

# 6 Conclusion

In this work, we have shown sample complexity improvements with lifelong learning in the challenging, yet as argued important setting, where tasks arrive in a stream (without assumptions on the tasks generating process), where the learner is only allowed to maintain limited amounts of information from previously encountered tasks, and where high performance is required for every single task, rather than on average. While such improvements have been established in very specific settings [3], our work shows they are possible in much more general and realistic scenarios. We hope that this will open the door for more work in this area of machine lifelong learning and lead to better understanding of how and when learning machines can benefit from past experience. An intriguing direction is to investigate whether there exists a more general characterization of ensemble methods and/or data distributions that would lead to benefits with lifelong learning. Another one is to better understand lifelong learning with neural networks, analyzing cases of more complex network structures and activation functions, an area where current machine learning practice yields exciting successes, but little is understood.

### Acknowledgments

This work was in parts funded by the European Research Council under the European Union's Seventh Framework Programme (FP7/2007-2013)/ERC grant agreement no 308036.

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
