[Supplementary Material]

# Lifelong Learning with Weighted Majority Votes: Supplementary Material

**Anastasia Pentina**
IST Austria
apentina@ist.ac.at

**Ruth Urner**
Max Planck Institute for Intelligent Systems
rurner@tuebingen.mpg.de

In this document we provide the proofs omitted from the main manuscript.

## 1 Properties of $d_D(\mathcal{H}, \mathcal{H}')$

**Claim 1.** *The distance $d_D(\mathcal{H}, \mathcal{H}') = \max_{h \in \mathcal{H}} d_D(h, \mathcal{H}')$ between two hypothesis sets satisfies the triangle inequality $d_D(\mathcal{H}_1, \mathcal{H}_3) \le d_D(\mathcal{H}_1, \mathcal{H}_2) + d_D(\mathcal{H}_2, \mathcal{H}_3)$.*

*Proof.*

for any $h_1 \in \mathcal{H}_1$:

$$
\begin{aligned}
d_D(h_1, \mathcal{H}_3) &= \min_{h_3 \in \mathcal{H}_3} d_D(h_1, h_3) \\
&\le \min_{h_3 \in \mathcal{H}_3} \left( d_D(h_1, h_2) + d_D(h_2, h_3) \right) \forall\, h_2 \in \mathcal{H}_2 \\
&= d_D(h_1, h_2) + \min_{h_3 \in \mathcal{H}_3} d_D(h_2, h_3) \forall\, h_2 \in \mathcal{H}_2 \\
&= d_D(h_1, h_2) + d_D(h_2, \mathcal{H}_3) \forall\, h_2 \in \mathcal{H}_2 \\
&\le d_D(h_1, h_2) + d_D(\mathcal{H}_2, \mathcal{H}_3) \forall\, h_2 \in \mathcal{H}_2
\end{aligned}
$$

by minimizing over $h_2$:

$$
d_D(h_1, \mathcal{H}_3) \le d_D(h_1, \mathcal{H}_2) + d_D(\mathcal{H}_2, \mathcal{H}_3)
$$

by maximizing over $h_1$ on the right hand side:

$$
d_D(h_1, \mathcal{H}_3) \le d_D(\mathcal{H}_1, \mathcal{H}_2) + d_D(\mathcal{H}_2, \mathcal{H}_3)
$$

by maximizing over $h_1$ on the left hand side:

$$
d_D(\mathcal{H}_1, \mathcal{H}_3) \le d_D(\mathcal{H}_1, \mathcal{H}_2) + d_D(\mathcal{H}_2, \mathcal{H}_3).
$$

$\square$

## 2 Proof of Lemma 2

We will prove the statement by induction on $k$ over a stronger statement that the conclusion holds for $V_k = \mathrm{MV}(w_1, \ldots, w_l, h_1, \ldots, h_k)$ and $\tilde{V}_k = \mathrm{MV}(w_1, \ldots, w_l, \tilde{h}_1, \ldots, \tilde{h}_k)$ for any $w_1, \ldots, w_l$. Note that for $k = 1$ the statement follows from Lemma 1.

Let $V_k' = \mathrm{MV}(w_1, \ldots, w_l, h_1, \ldots, h_{k-1}, \tilde{h}_k)$. Then:

$$
\begin{aligned}
d_D(V_k, \tilde{V}_k) &\le d_D(V_k, V_k') + d_D(V_k', \tilde{V}_k) \ \ \text{(by triangle inequality)} \\
&\le d_D(h_k, \tilde{h}_k) + d_D(V_k', \tilde{V}_k) \ \ \text{(by Lemma 1)} \\
&\le \epsilon_k + \sum_{i=1}^{k-1} \epsilon_i \ \ \text{(by assumption and induction)}.
\end{aligned}
$$

# 3   Proof of Theorem 3

1. First, as in the proof of Theorem 2, we need to control the total probability of any conclusion of Algorithm 2 being incorrect. For every task $i = 2, \ldots, n$ Algorithm 2 preforms at most two estimations. Therefore the total probability of failure is:

$$\delta_1 + \sum_{i=2}^{n} 2\delta_i = \frac{\delta}{2} + \sum_{l=1}^{\lfloor \log n \rfloor} 2(2^{l+1} - 2^l)\frac{\delta}{2^{2l+2}} = \frac{\delta}{2} + \frac{\delta}{2} \sum_{l=1}^{\lfloor \log n \rfloor} \frac{1}{2^l} \le \frac{\delta}{2} + \frac{\delta}{2} \sum_{l=1}^{\infty} \frac{1}{2^l} = \frac{\delta}{2} + \frac{\delta}{2} = \delta.$$

2. Performance guarantees follow from the design of the algorithm (as in Theorem 2).

3. The fact that $\tilde{k} \le k$ can be proven in a way analogous to Theorem 2. However, we need to make sure that for every $\hat{k} = 1, \ldots, \tilde{k}$, by using Lemma 2, we will obtain a suitable result. In particular, by construction for every $j = 1, \ldots, \hat{k} - 1$ $d_{D_{i_j}}(h_{i_j}^*, \tilde{h}_j) \le \epsilon_j'$. Therefore by Lemma 2:

$$d_{D_{i_{\hat{k}}}}(MV(h_{i_1}^*, \ldots, h_{i_{\hat{k}-1}}^*), MV(\tilde{h}_1, \ldots, \tilde{h}_{\hat{k}-1})) \le (\hat{k} - 1)\xi + \sum_{j=1}^{\hat{k}-1} \epsilon_j'. \tag{1}$$

By the definition of $\epsilon_j'$:

$$\sum_{j=1}^{\hat{k}-1} \epsilon_j' \le \frac{\epsilon}{16} + \sum_{m=1}^{\lfloor \hat{k} \rfloor} (2^{m+1} - 2^m)\frac{\epsilon}{2^{2m+4}} = \frac{\epsilon}{16} + \frac{\epsilon}{16} \sum_{m=1}^{\lfloor \hat{k} \rfloor} \frac{1}{2^m} < \frac{\epsilon}{16} + \frac{\epsilon}{16} = \frac{\epsilon}{8}.$$

Together with the assumption on discrepancies, this guarantees that:

$$d_{D_{i_{\hat{k}}}}(\mathrm{MV}(h_{i_1}^*, \ldots, h_{i_{\hat{k}-1}}^*), \mathrm{MV}(\tilde{h}_1, \ldots, \tilde{h}_{\hat{k}-1})) \le \frac{\epsilon}{4}, \tag{2}$$

which is exactly what is needed to come to contradiction.

4. The sample complexity of Algorithm 2 consists of the same parts as that of Algorithm 1.

The first difference comes from the fact that $\delta'$ changes over time, because the algorithm does not know the total number of tasks. However, the smallest value it attains is $\delta/(4n^2)$ and, since the dependence of the sample complexity on the $\delta$ is only logarithmic, it does not change the result significantly.

The second difference is that also $\epsilon'$ changes over time, because the algorithm does not know the parameter $k$ in advance. This influences the sample complexity of learning "base tasks". In order to control it we need to control the following sum:

$$\sum_{j=1}^{\tilde{k}} \frac{1}{\epsilon_j'} \le \sum_{m=1}^{\lfloor \log k \rfloor} (2^{m+1} - 2^m)\frac{2^{2m+4}}{\epsilon} = \frac{16}{\epsilon} \sum_{m=1}^{\lfloor \log k \rfloor} 2^{3m} \le \frac{k^3 \log k}{\epsilon}.$$

Therefore the complexity of learning the "base tasks" is:

$$\tilde{O}\left(\frac{\mathrm{VC}(\mathcal{H})k^3}{\epsilon}\right). \tag{3}$$