[Reviews · NeurIPS 2016]

Reviewer 1

Summary

The paper presents a learning algorithm that can learn many (different but related) prediction tasks by transferring knowledge obtained from previously solved tasks. The algorithm receives the tasks sequentially and choose to either: - build a majority vote predictor by weighting previously learned predictors; or - learn a new predictor "from scratch". The algorithm is inspired by VC-dimension regret bounds and is designed to obtain theoretical generalization and sample complexity guarantees. However, no empirical experiments are performed.

Qualitative Assessment

From my very personal point of view, the lifelong learning paradigm is a vague concept and is sometimes evoked for studying different scenarios that can be named otherwise (like transfer learning). However, I think that the framework studied here (from Balcan et al., 2015) is a very pertinent "lifelong problem". The authors present an honest work in the right direction. The proofs are not trivial but, as I explain below, the contribution appears to me insufficient for NIPS. The risk bound minimized by the learning algorithm may be very high, as it relies on the VC-dimension of the predictors. Also, the role of the constant C in Theorem 1 is never discussed, and little is said about the way to handle it in the algorithm. Therefore, I am unconvinced that the algorithm will lead to a successful empirical procedure. I consider that the authors should provide empirical evidences that the approach is sound (at least on a synthetic dataset). It is also likely that a tighter bounding technique (such as Rademacher or PAC-Bayes, both used by some of the cited works) provides better theoretical guarantees and/or empirical results. If there is a particular reason to rely on VC-dimension, the authors should discuss their choice. In the introduction (Lines 47 to 50, and 80-81), the authors claim that, as opposed to Balcan et al. (2015), their algorithm learns "with weighted majority votes rather than linear combinations over linear predictor" and that "the shift from linear combinations to majority votes introduces a stability to the learned ensemble that allows exploiting it for later tasks." However, this comparison is not explicitly discussed later on. The meaning of the claim is unclear to me; In general, a linear combination of predictors can be expressed as a majority vote with a simple reparameterization. *** Post-rebuttal comments: Considering the authors’ rebuttal, I raised my "technical quality" score a bit. I still think that the paper would benefit from some experiments, in order to assess if Algorithm 1 would make an empirically good learning algorithm. It would also help to answer practical questions like: - Is the "\gamma-effective dimension" assumption realistic? - Does the number of required labels for each task will be reasonable? Or should we manually "override' the constant C to make the algorithm work empirically? The authors argued that we must consider the paper as a theoretical work, and leave these questions for future work. This is defendable. Personally, I feel that a work presenting a learning algorithm without any empirical results (not even on a toy dataset) is a bit incomplete (sure, Balcan et al. did it beforehand, but in COLT, which is strongly theory oriented). Finally, regarding the linear combinations vs majority votes misunderstanding, I think the distinction must be clearly specified in the paper. From my point of view, a linear *separator* is given by \sign( < v_i, x > ), albeit a linear *regressor* is given by < v_i, x >. Moreover, a majority vote ca be a aggregation of either classifiers or regressors.

Confidence in this Review

2-Confident (read it all; understood it all reasonably well)


Reviewer 2

Summary

The authors consider lifelong learning using weighted majority votes of functions that previously had to be learned from the original hypothesis space. The algorithm first tries to learn a newly encountered function from the weighted majority vote. If it fails, it tries to learn from the original hypothesis space. When the number of functions that cannot be well approximated by the weighted majority is small, the sample complexity of the lifelong learning method is correspondingly shown to be small compared to learning all the functions from the original hypothesis space.

Qualitative Assessment

The authors show that to learn the weighted majority of previously learned functions, you only need to learn the weighted majority of functions where learning previously failed, assuming that the functions to be learned all come from the same base class and the tasks have small discrepancy in the marginal distributions. This allows life-long learning to be more sample (as well as other resource) efficient. The authors introduce the $\gamma$-effective dimension, which bounds the number of failures. I think that the result shows that weighted majority of previously learned functions may potentially be useful for lifelong learning. However, the extent of its usefulness is unclear. In particular: - Are there base function classes with small $\gamma$-effective dimension (for the worst case sequence)? - How do we compute the $\gamma$-effective dimension so that we can tell whether a class H is suitable to use with weighted majority for lifelong learning? - I suppose the ordering of the task matters. Would be good to see some discussion on that. Experimental results would also be useful to see the performance of weighted majority for lifelong learning. Using a good example in the writing would make the paper more accessible to a wider audience. For example, learning a linear classifier from image features for object classification for a very large number of classes (where each class is a task) from the same distribution of images (same distribution for each task, so zero discrepancy).

Confidence in this Review

2-Confident (read it all; understood it all reasonably well)


Reviewer 3

Summary

The paper analyzes a streaming model of lifelong learning in which the learner can only retain the learned classifier from previous tasks. This is a significantly harder setting for lifelong learning than those studied previously in the literature. The paper presents a general learning algorithm that uses majority votes of base learners to learn tasks in this setting. It also introduces a new notion of dimension for sequences of learning tasks which bounds the sample complexity of the presented learning algorithm. This sample complexity can be significantly smaller than the sample complexity of the naive algorithm that learns each task independently.

Qualitative Assessment

Using representations learned from previous tasks to be more efficient at new learning tasks is an important problem in machine learning, both from a theoretical and a practical perspective. The model introduced in this paper seems to be cleaner and perhaps more realistic than lifelong learning models in the previous literature. The proposed algorithm maintains a set of majority votes over the base hypotheses, uses already learned majority votes when they have low enough empirical risk, and learns a new weighted majority vote and adds them to the set otherwise. As the authors remark, this is a very natural paradigm and similar uses of majority votes in learning algorithms are fairly standard in the literature. Nevertheless, the analysis is nontrivial and the presentation is elegant and clear. On the other hand, more context for the newly introduced dimension and the sample complexity bounds would have been helpful. For one, the γ-effective dimension introduced here, unlike most classical dimensions that characterize learnability in various models, depends on the distributions on the data. If such a dependence is necessary, an example demonstrating how worst-case distributions can make lifelong learning impossible would help the reader understand the model. Some context and examples for the sample complexity bounds would help, too. Although it is clear from the statement of the theorem that, if the γ-effective dimension is small enough, we can get a big sample complexity improvement, it is less clear what kind of learning tasks can be expected to realize this. Also, it would be interesting to understand how much further improvement we could hope for in this model. Even if the authors have been unable to prove any sample complexity lower bounds that would give us this context, conjectures or some discussion about the true sample complexity would be interesting. It seems that the use of ``min’’ in the definition of distance, and the claim (11) require justification. Overall, this is a well-written paper that makes a significant contribution to an important field of machine learning.

Confidence in this Review

2-Confident (read it all; understood it all reasonably well)


Reviewer 4

Summary

This paper attempts to improve sample complexity with lifelong learning in the setting where tasks arrive in a stream.

Qualitative Assessment

The algorithm proposed in this paper is similar to the algorithm of stacking, which trains multiple weak classifiers and then learns a weighted combination of the outputs of these weak classifiers. The most important issue of stacking is overfitting, so I think there is the same issue in the algorithm proposed in this paper, since the hypothesis learnt for the i-th task heavily depends on the tasks from 1 to i-1. Some experimental results should be conducted to show the effectiveness of the proposed algorithm. There is no significant contribution in the theoretical part, it is a straightforward result with the VC dimension of the weighted combination. I do not understand why the section 4.2 is necessary in this paper, this paper has little to do with neural networks.

Confidence in this Review

2-Confident (read it all; understood it all reasonably well)


Reviewer 5

Summary

This paper proposed a lifelong learning algorithm based on weighted majority vote and provided performance analysis. From a general view, the algorithm learns a binary classifier by using weighted majority vote on previous classifiers with sufficient large labeled data for each sequential task. If the learned classifier cannot reduced the error rate to certain level, then it will re-train an classifier from the ground base classifier set.

Qualitative Assessment

The idea of using weighted majority votes in lifelong learning setting is novel. In the analysis part, this paper provides guarantee on maximum error rate for each task instead of expected error rate compared with previous literature cited in the paper. The authors also draw some interesting connections between the proposed learning algorithm with neural network with rectifier activation function. One question I have is that in both Alg.1 and Alg.2, there are critical steps to draw training set which depend on an condition on $\Delta$, which is further depend on an unknown parameter C. How can resolve this issue in practice?

Confidence in this Review

1-Less confident (might not have understood significant parts)


Reviewer 6

Summary

The paper studies the problem of lifelong learning with "safety" constraints, where a learner needs to solve a sequence of tasks (up to some accuracy), but are not allowed to keep previous training data as it attempts to solve a new task. The authors make no distribution assumption on the tasks. They show that if the tasks are related (characterized by a notion called the "effective dimension"), then in theory, one can achieve a significant reduction in sampling complexity by transferring information from previous tasks, as opposed to learn each individual task in isolation.

Qualitative Assessment

Pros: The paper is clearly written and easily understandable. It provides an algorithmic framework for lifelong learning task, including strategies for both the case when the number of tasks and base solvers are known, and the case when such information is not available. For for the latter the authors use a doubling trick, which adaptively chooses the model parameters on the fly. The inductive proof is sound and explained well; it is interesting to see the authors' view on its connection with learning representations in a neural network. Cons: My main concern is the lack of empirical evidence, to show that why majority votes are superior to linear combination of hypotheses. Since the paper suggests a new algorithmic tool and claim it is suitable for "realistic" scenarios, it would have made the claim much stronger if the authors can design experiments to conduct some (at least preliminary) empirical evaluation. While the theoretical propositions in the paper are mostly clearly explained, intuitively, can the authors please explain what is the main advantage of majority vote hypothesis class, over the linear combinations of bases linear predictors? Why does it offer robustness behavior in comparison with linear combinations? A high-level explanation will be helpful to make the paper clearer.

Confidence in this Review

1-Less confident (might not have understood significant parts)